# Fano resonant optical coatings platform for full gamut and high purity structural colors

Mohamed ElKabbash[1,4,5] ✉, Nathaniel Hoffman [2,5], Andrew R. Lininger [2,5], Sohail A. Jalil[1], Theodore Letsou[2], Michael Hinczewski [2] ✉, Giuseppe Strangi [2,3] ✉ & Chunlei Guo[1] ✉

Structural coloring is a photostable and environmentally friendly coloring approach that harnesses optical interference and nanophotonic resonances to obtain colors with a range of applications including display technologies, colorful solar panels, steganography, décor, data storage, and anticounterfeiting measures. We show that optical coatings exhibiting the photonic Fano Resonance present an ideal platform for structural coloring; they provide full color access, high color purity, high brightness, controlled iridescence, and scalable manufacturing. We show that an additional oxide film deposited on Fano resonant optical coatings (FROCs) increases the color purity (up to 99%) and color gamut coverage range of FROCs to 61% of the CIE color space. For wide-area structural coloring applications, FROCs have a significant advantage over existing structural coloring schemes.

In nature, colors are mostly produced either through pigments or structures. While the former comes from molecular absorption, the latter, structural coloring, is the result of optical interference from a structured surface. Structural colors offer several advantages over pigments; they are photostable, immune to chemical degradation, and environmentally friendly. In addition, a wide range of colors can be produced[1] and dynamically reconfigured[2] using the same material. Furthermore, structured surfaces can have multiple functionalities, e.g., creating hydrophobic or antibacterial colored metals[3,4].

Several structural coloring schemes have been previously introduced including multilayer films[5], thin film nanocavities[6], plasmonic nanostructures[1], dielectric nanostructures[7], and photonic crystals[8] with applications in decoration[9], colorimetric sensing[10], data storage[11], anticounterfeiting[12], display technologies[13], colorful photovoltaic cells[14], among others[15]. An ideal structural coloring platform should satisfy several requirements. More specifically, it should span the widest possible area of the color gamut with high and controllable purity, i.e., how monochromatic the color is, and brightness, i.e., the relative intensity of the reflected color. The platform should also allow control over the iridescence, i.e., the angle dependence of observed

colors. Some iridescence is common for structural colors generated via interference. For many applications, structural coloring platforms should be scalable and inexpensive to fabricate. However, no existing scheme has been shown to satisfy all the above qualities simultaneously[1,7,9,14].

Recently, we proposed a new type of optical coatings that exhibits the photonic Fano Resonance effect[16]. Fano Resonant Optical Coatings (FROCs) enjoy unique optical properties that cannot be reproduced with existing optical coatings such as metallic films, anti-reflective coatings, transmission filters, light absorbers, and dielectric mirrors. Figure 1a describes the composition of FROCs. FROCs are produced by coupling a broadband nanocavity[17] (representing the continuum) with a narrowband Fabry-Perot (FP) nanocavity[18] (representing a discrete state). The resonant interference between the nanocavities produces the well-known asymmetric Fano resonance line-shape.

In this work, we develop a class of reflective FROCs by using a reflective and opaque material as substrates, which lead to a highly reflective resonant peak that corresponds to the narrowband nanocavity's resonance. We investigate the color properties of FROCs numerically and experimentally. We show that the reflective FROCs are

[1]The Institute of Optics, University of Rochester, Rochester, NY 14627, USA. [2]Department of Physics, Case Western Reserve University, 10600 Euclid Avenue, Cleveland, Ohio 44106, USA. [3]CNR-NANOTEC and the Department of Physics University of Calabria, Rende, Italy. [4]Present address: College of Optical Sciences, University of Arizona, Tucson, AZ 85721, USA. [5]These authors contributed equally: Mohamed ElKabbash, Nathaniel Hoffman, Andrew R. Lininger. ✉e-mail: melkabbash@arizona.edu; mxh605@case.edu; gxs284@case.edu; guo@optics.rochester.edu

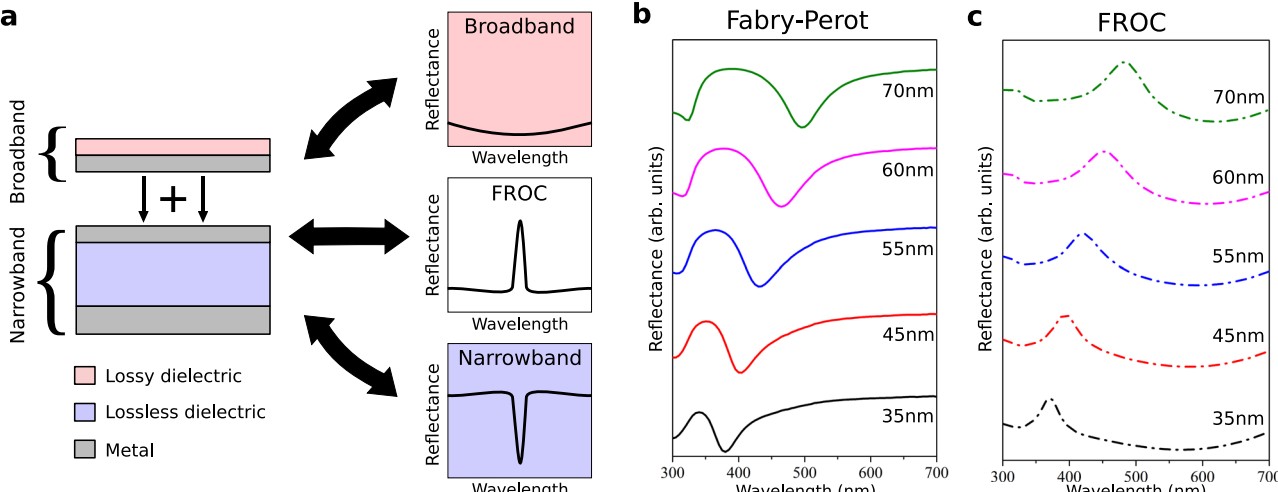

**Fig. 1 | Spectral properties of fano resonant optical coatings. a** A FROC consists of two coupled light absorbers; a broadband absorber and a narrowband (Fabry-Perot, FP) absorber. A FROC exhibits a reflection peak at the FP cavity resonance. **b** The measured reflection from a FP cavity with different dielectric thickness. **c** The measured reflection from the same FP cavities shown in (**b**) after depositing a 15 nm Ge film to create a FROC. The incidence angle in (**b**) and (**c**) is 15°.

an ideal platform for structural coloring, producing colors spanning a wide color gamut with high brightness and high purity. The purity and color gamut coverage are further enhanced by introducing a dielectric thin film capping layer. The dependence of the color on the incident angle can be controlled through the dielectric cavity material, making FROCs suitable for a variety of applications that demand angle independent, e.g., decoration, or angle dependent coloring, e.g., anti-counterfeit measures. Structural coloring with FROCs can find new applications where strong and broadband optical absorption and high purity colors are required, for example, colorful solar thermal generation panels, colorful photovoltaic panels[8], colorful thermo-photovoltaic panels, and colorful solar thermoelectric generators. These renewable energy sources often have the same color, black or dark blue, which makes them esthetically unappealing.

## Results

Throughout this work, we compare the coloring performance of FROCs to FP thin film nanocavities since they are the closest platform in terms of structure and physics to FROCs[16]. FP nanocavities produce colors through selective absorption, mainly reflecting all colors except for the specific cavity resonance wavelength. On the other hand, FROCs produce colors through selective reflection (Fig. 1a). Figure 1b shows the measured p-polarized reflection spectrum of FP nanocavities consisting of Ag (20 nm)-TiO₂-Ag (100 nm) by varying the thickness of the TiO₂ film from 35 nm to 70 nm (Fig. S1). Figure 1c shows the measured reflection spectrum of the same FP nanocavities after adding a 15 nm Ge layer to convert them into FROCs. The reflection band produced by the FROCs correspond to the absorption band produced by the FP cavity. Consequently, by changing the dielectric thickness/refractive index of the FP cavity, FROCs can reflect a relatively narrow range of wavelengths that can be tuned across the visible spectrum.

To examine the structural coloring properties of FROCs vs. FP cavities, we calculate the colors produced from FP cavities vs. FROCs by varying the top metal film thickness and the cavity thickness. Figure 2a shows a swatch array for FP cavities consisting of a metal-dielectric-metal stack [top to bottom: Ag (Y nm)- TiO₂ (X nm)- Ag (100 nm)] as a function of the top Ag film thickness and the TiO₂ dielectric cavity thickness. The colors in the swatch array are the perceived color for a person viewing the sample. The produced colors are Cyan-Magenta-Yellow (CMY) colors since FP cavities are selective absorbers and the CMY model describes subtractive colors[19]. Note the

restricted color palette from FP nanocavities. Figure 2b shows the corresponding purity of the FP cavities (see Methods for more details). The purity of FP cavities is limited since all the colors are reflected except within the absorbed wavelength range, making near-monochromatic reflection impossible. Figure 2c shows a swatch array of FROCs utilizing the same FP cavities with an added absorbing thin film [Ge (15 nm)] to generate the desired Fano resonance. A wide range of colors can be seen in the swatch array. Since FROCs are selective reflectors, these colors are RGB as opposed to CMY. This allows for high purity color reflection (near monochromatic), as illustrated by the purity array in Fig. 2d. FROCs exhibit overall greater purity colors than the FP cavities, with near maximal purity for some configurations.

However, close inspection of Fig. 2c and Fig. 2d reveals that high purity green and red colors are still difficult to obtain. This is because pure red and green colors require having the Fano resonance peak at longer wavelengths. On the other hand, the measured reflection from the Ge-Ag broadband absorber is >0.15 at short wavelengths <500 nm (Fig. 2g). Consequently, the color purity drops since shorter wavelengths are strongly reflected. To suppress the high reflection at shorter wavelengths, we add a 50 nm silicon dioxide (SiO₂) capping layer Fig. 2g. Figure 2h shows the measured reflection for a FROC that produces a green color with and without the SiO₂ capping layer. The SiO₂-FROCs exhibit significantly reduced reflection at short wavelengths and relatively small change near the resonance peak. This combination leads to an overall increase in the colorimetric purity. Figure 2e and Fig. 2f show the swatch array and corresponding color purity of the SiO₂-FROCs design, respectively. By comparing Fig. 2e, f, with Fig. 2c, d, respectively, SiO₂-FROCs are shown to produce greater purities compared to the original FROC structure. A purity level > 99% can be reached (Fig. S2) which is higher than other thin-film structural color platforms[7]. We note that any lossless dielectric other than SiO₂ can act as a capping layer which is meant to act as an impedance matching layer that further suppresses the reflection from the broadband absorber (see Fig. 2g, h).

Images of the reflected color for experimentally fabricated FROCs and FP cavities with a TiO₂ thickness ranging from 35 nm to 150 nm are shown in Fig. 3a, b, respectively. FROCs are capable of reflecting blue, green, and red colors by simply increasing the dielectric cavity thickness. SiO₂ capped FROCs further boost the color purity as shown in Fig. 3c for blue, green, and red colors (see also Fig. S5). Figure 3d shows a photograph of two FP cavities with CWRU and U of R letters printed

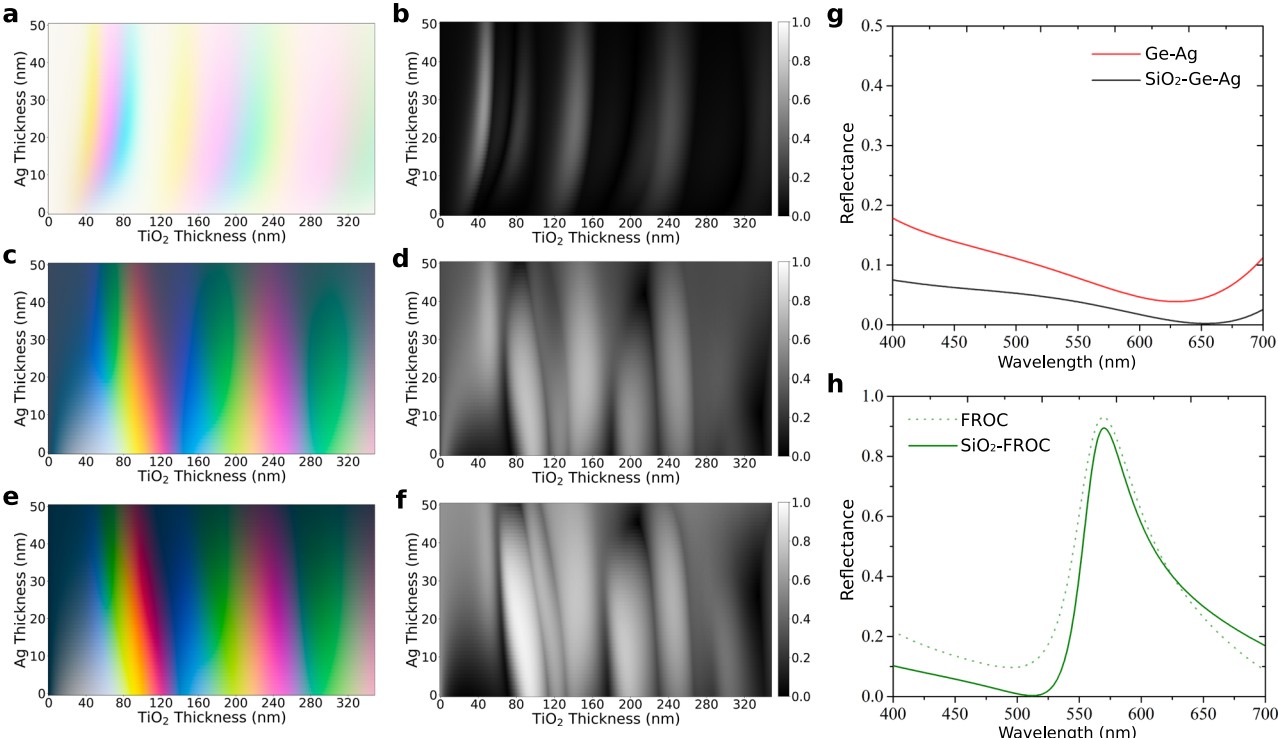

**Fig. 2 | Colorimetric properties of FROCs: Swatch array and the corresponding color purity for Fabry-Perot (FP) nanocavities. a, b** FROCs (**c**) and (**d**) and Silica-capped FROCs (**e**) and (**f**). **g** Measured reflectance of a Ge-Ag broadband absorber (red line) vs. a silica (50 nm) capped broadband absorber (black line). Adding a silica film reduces the overall reflectance from the broadband absorber. **h** Measured reflectance of a FROC with a Fano resonance peak within the green wavelength range with and without a silica cap. The suppressed reflectance at shorter wavelengths is evident.

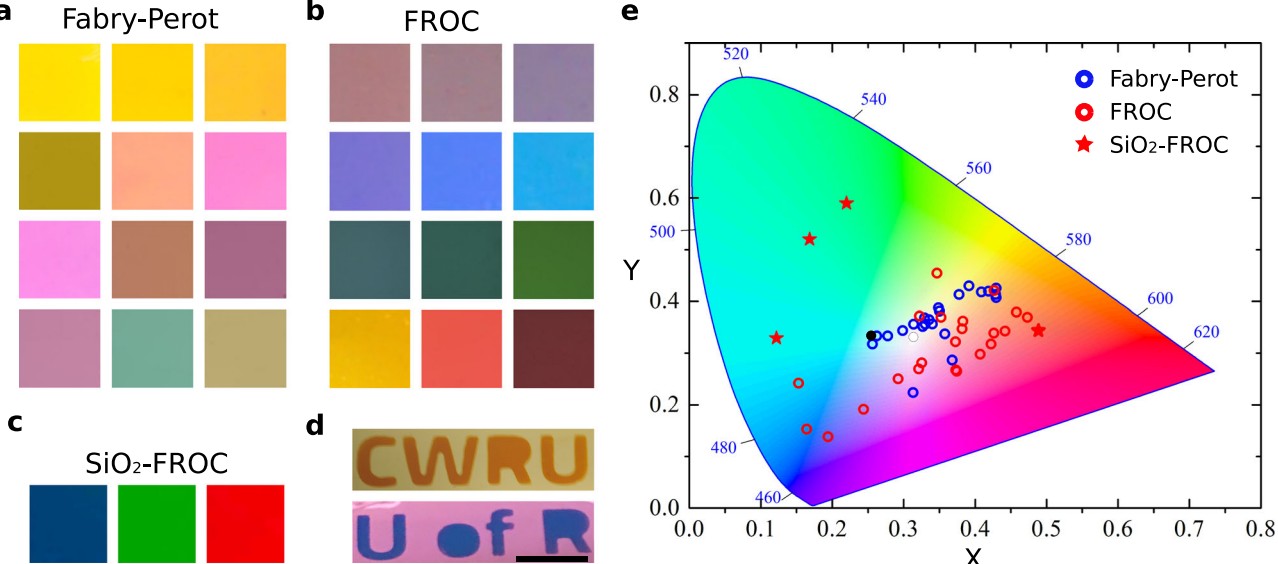

**Fig. 3 | Structural coloring with FROCs. a, b** Show photographs of fabricated FP nano-cavities and FROCs. **c** Shows photographs of blue, red, and green colors generated using the SiO₂ capped FROC system. The color purity of FROCs is evident in (**d**) where the letters of U of R and CWRU are printed on a FP cavity by depositing 15 nm Ge layer. Scale bar = 1 cm. **e** The CIE 1931 color space showing the colors corresponding to the measured reflection spectrum of FP nano-cavities (blue circles) and FROCs (red circles). FROCs demonstrate higher purity as they are further away from the white point (black dot). Silica capped FROCs (red stars) show higher purity in the green region of the color space.

on them by depositing a 15 nm Ge layer and converting these regions to a FROC. By controlling the spatial distribution of the deposited layer, this printing method could be adapted for optical archival data storage and encrypting messages. Experimentally measured reflectance from different FP cavities (blue dots) and FROCs (red dots) are presented in the CIE 1931 color space (see Methods) (Fig. 3e). The white point (black dot) corresponds to the spectrum of the illuminant, i.e., white light. Note that the color in the photographs may appear different due to the spectrum of the illuminant. The selective reflection from FROCs enables access to a wide range of colors from blue to red, including high purity green. The colors enjoy significantly high purity[20].

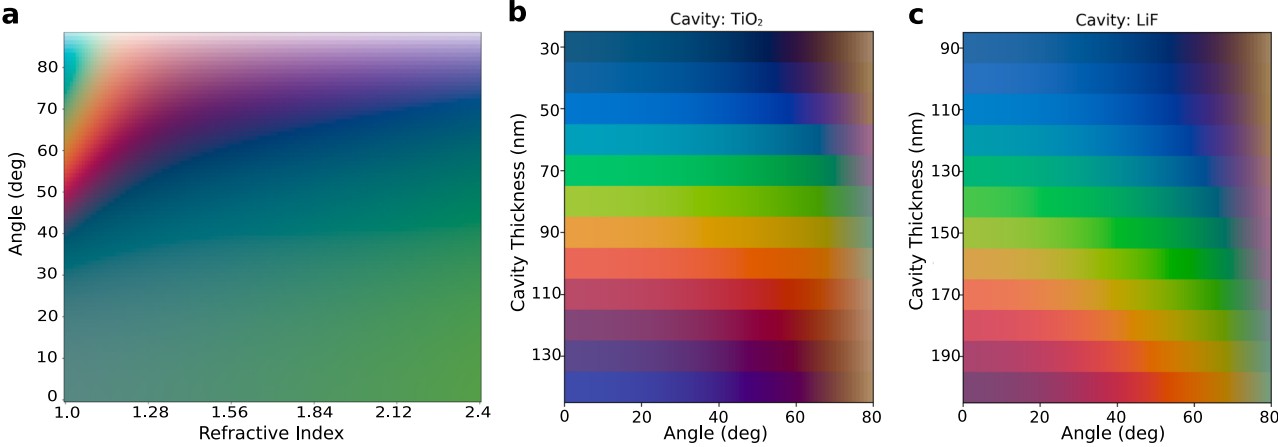

**Fig. 4 | Angle dependence of FROCs. a** Swatch array of a silica capped FROC for a constant optical length by varying the refractive index and incidence angle. The angle dependence can be controlled by the refractive index of the dielectric cavity. **b, c** Show a swatch array as a function of the cavity thickness and incidence angle for a FROC with real dispersive materials TiO$_2$ ($n \sim 2.3$) and LiF ($n \sim 1.4$) as the dielectric cavity material.

The iridescence properties of FROCs can be controlled by simply changing the refractive index of the dielectric cavity material. Note that tunable iridescence is a desired property in many applications, e.g., anti-counterfeit measures[12,21]. Figure 4a shows a swatch array (angle of incidence vs. dielectric cavity material refractive index) for a range of FROCs with constant dielectric cavity optical length $l = n\,t = 700\,nm$, where $n$ is the dielectric index and $t$ is its thickness.. At high refractive index FROCs exhibit low iridescence (relatively unchanged color reflectance) over a wide angular range ($0° - 70°$). At low refractive index the iridescence increases, implying a tunable control over the iridescence properties. This analysis assumes that the dielectric cavity material is lossless and dispersionless for simplicity. However, introducing typical losses to the cavity material does not affect the general behavior captured in Fig. 4a (Fig. S7 and Fig. S8). Similarly, the iridescence properties are qualitatively similar when introducing dispersion to the cavity material (Fig. S6).

Figures 4b and 4c present swatch arrays (angle of incidence vs. dielectric cavity thickness) for FROCs with real dielectric cavity material dispersions: a high-index (TiO$_2$) and a low index (LiF) dielectric. Controllable iridescence is demonstrated for a wide color range spanning the CIE gamut. This includes low iridescence with the high index dielectric cavity material.

Finally, we assess the color range covered by silica capped FROCs. FROCs access a significantly larger color gamut since the wavelength of the selective reflection is tuned by simply changing the dielectric thickness. Figure 5a shows the CIE 1931 chromaticity coordinates of FROCs vs. FP cavities. The CIE gamut coverage for both approaches is maximized by employing a least-squares based optimization procedure to optimize the individual layer thicknesses to produce maximal purity (Fig. S9, and related text)[22]. FP cavities are considered in two varieties: tuning the thickness of all three layers in the metal-dielectric-metal stack (all layers) or assuming the bottom metal layer is maximally reflective and opaque (opaque base). Although opaque base FP cavities are the most direct comparison with FROCs, a larger color space can be accessed by optimizing all three layers. At normal incidence FROCs cover 43.9% of the CIE gamut, while SiO$_2$-FROCs cover 61.3%. This includes a near complete (>99%) coverage of the sRGB and Adobe RGB color spaces with a total covered area reaching 131.6% and 177.5% compared to the Adobe RGB and sRGB color subspaces, respectively (Fig. S10). On the other hand, FP cavities only cover 31.8% of the CIE color space even when optimizing all three layers. FROCs are also compared with several of the best performing thin-film structural coloring platforms in the recent literature (Fig. 5a). The color response

of these platforms is simulated based on the structures reported in the literature, but not further optimized. To the best of our knowledge, the CIE gamut coverage of FROCs is larger than any other thin-film structural coloring platform[19,23–27] (Table S1). In fact, optimized SiO$_2$-FROCs can cover a larger color gamut compared to the state-of-the-art 2D nanostructure-based structural coloring approaches[7,28].

Another important metric is brightness, i.e., the reflected intensity compared to the incident light at the peak resonance wavelength. The calculated peak reflectance from FROCs is >0.9 (Fig. S11) and the measured reflectance ranges from 0.63- 0.85 as shown in Fig. 5c. The lower reflectance in the measured films can be improved by depositing higher quality silver films and precisely controlling thickness[29].

## Discussion

Figure 5d compares the structural coloring performance metrics of FROCs, dielectric nanostructures, plasmonic nanostructures, and photonic crystals. FROCs outperform existing structural coloring methods by providing high purity and color gamut coverage, high brightness, angular control, and inexpensive and scalable fabrication[1]. Because thin film structural coloring generally has lower spatial resolution compared to 2D nanostructure-based structural coloring, the latter approach remains advantageous for high density coloring, although FROCs are preferential for large area coverage. Durability is a common issue for applications of optical coatings. Durable FROCs could be made using ceramic materials[30]. We believe that FROCs are particularly suitable for colored solar thermal panels as they are efficient in absorbing the solar spectrum[16] while reflecting an on demand, narrowband color. In addition, by using amorphous Si instead of Ge and utilizing the metal films in the FROC to act as electrodes, it is possible to realize colorful photovoltaic cells[14] using the well-established thin film deposition technologies.

## Methods

### Sample fabrication

The FROC films were deposited on a glass substrate (Micro slides, Corning) using electron-beam evaporation for Ge (3 Å s$^{-1}$) and TiO$_2$ (1 Å s$^{-1}$) pellets and thermal evaporation Ag (20 Å s$^{-1}$), with the deposition rates specified for each material. The silica capped FROC films were deposited on a glass substrate (2948, Corning) using electron-beam evaporation for Ge (0.5 Å s$^{-1}$), TiO$_2$ (1 Å s$^{-1}$), and SiO$_2$ (0.8 Å s$^{-1}$), and DC magnetron sputtering for Ag (2 Å s$^{-1}$). All deposition materials were purchased from Kurt J. Lesker. Deposited layer thicknesses were measured with spectroscopic ellipsometry (J. A. Woollam).

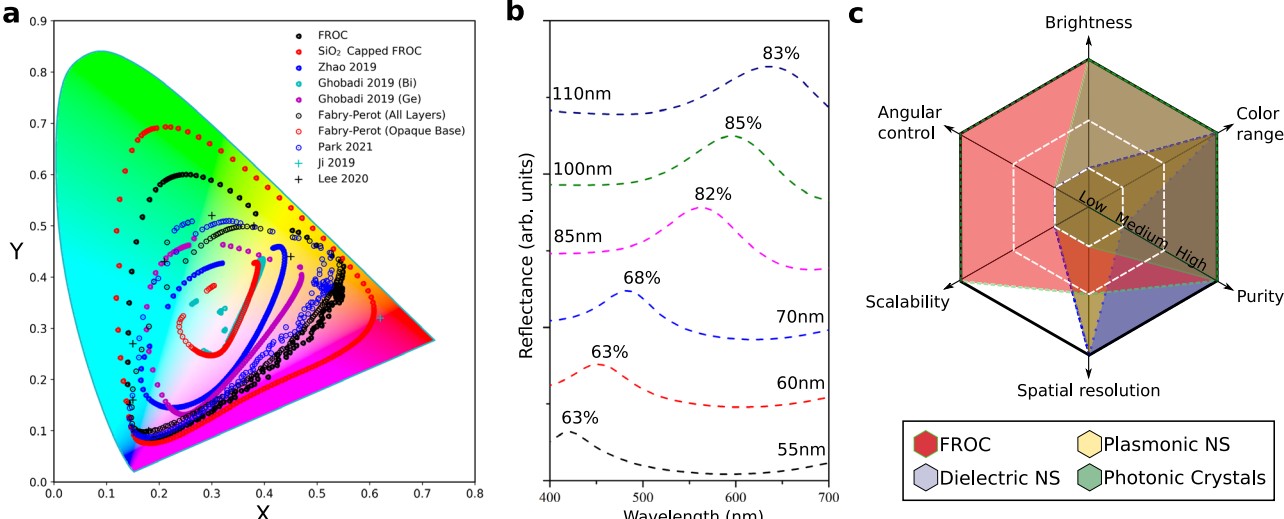

**Fig. 5 | Accessible color gamut, brightness, and overall performance of FROCs.**
**a** CIE 1931 chromaticity diagram comparing FROCs and Silica-capped FROCs with Fabry-Perot cavities and other reported thin-film based coloring approaches from the recent literature at normal incidence. The incidence angle is 0 °. **b** Measured reflectance from different FROCs showing a peak reflectance ranging from 0.63 to 0.85. The high reflectance corresponds to high brightness, a desirable property in structural colors. **c** Comparison between the coloring performance of FROC vs. other structural coloring platforms.

## Numerical calculation of the reflection and absorption spectra
Numerical reflectance and absorbance spectra were generated using a transfer matrix-based simulation model written in Mathematica and Python. Spectral optical constants for the multiple materials were obtained variously from the Brendel-Bormann model (Ag), fits to the experimental materials ($SiO_2$, $TiO_2$), or an amorphous experimental model for Ge. Transmittance was zero for all structures. Absorbance was calculated as the complementary to reflectance, or $A = 1 - R$.

Reflection measurements: Experimental angular reflectance measurements were performed using a variable-angle high-resolution spectroscopic ellipsometer (V-VASE, J. A. Woollam). Sample transmittance was zero for all angles and wavelengths. Experimental color profiles were analyzed as described in the following section. The colors panels shown in Fig. 3a–d were obtained with a standard camera under F11-type illumination.

## Color analysis
The reflectance spectra to CIE 1931-xyz colorspace conversation was performed in Python utilizing interpolations of the standard observer distributions[12,22]. The XYZ tristimuslus values are given as:

$$X = \frac{\int_\lambda S(\lambda)\alpha(\lambda)R(\lambda)d\lambda}{\int_\lambda S(\lambda)\beta(\lambda)d\lambda}, Y = \frac{\int_\lambda S(\lambda)\beta(\lambda)R(\lambda)d\lambda}{\int_\lambda S(\lambda)\beta(\lambda)d\lambda}, and\ Z = \frac{\int_\lambda S(\lambda)\gamma(\lambda)R(\lambda)d\lambda}{\int_\lambda S(\lambda)\beta(\lambda)d\lambda}$$

(1)

with $S$ as the illuminant spectrum, $R$ as the spectral reflectance, $k$ as a constant factor, and $\alpha, \beta, \gamma$ as the standard observer functions. The integration is over the visible spectrum. The CIE 1931-xyz values are then given as:

$$x = Y/N, y = Y/N, and\ z = 1 - x - y$$

(2)

for $N = X + Y + Z$. Note that at constant luminance, the chromaticity is defined by $x$ and $y$.

Color swatch arrays were generated by transforming the calculated chromaticity values into their sRGB equivalents using a matrix transform calculated from reference primaries with the D65 reference white and sRGB companding (IEC 61966-2-1 standard). Colors were generated with matplotlib in Python.

Excitation purity is calculated as:

$$p = |s - w|/|d - w|$$

(3)

where $s$, $w$, and $d$ are the CIE 1931 (x, y) coordinates for the measured spectra point, white point, and dominant wavelength point, respectively.

Total CIE x-y space coverage is calculated as the area of the smallest convex hull encompassing all of the desired CIE (x, y) points. Area is presented relative to the area of the full visible light color gamut (maximum purity). Sufficient resolution was obtained in the numerical simulations to approach a smooth hull.

## Data availability
Data used in this study are available from the corresponding authors upon request. The experimental and comparison data are provided in the Source Data file. Source data are provided with this paper.

## Code availability
The simulation and optimization code are freely available on GitHub at this link: https://github.com/hincz-lab/structural_color_FROCs[31].

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

## Acknowledgements

C.G. acknowledges the support of the Army Research Office and The Bill and Melinda Gates foundation (INV-009181). A.R.L., G.S., and M.H. acknowledge the NSF Grant 1904592, "Instrument Development: Multiplex Sensory Interfaces Between Photonic Nanostructures and Thin Film Ionic Liquids.". The authors acknowledge the use of the Materials for Optoelectronics Research and Education (MORE) Center, a core facility at Case Western Reserve University (est. 2011 via Ohio Third Frontier grant TECH 09-021). This work made use of the High-Performance Computing Resource in the Core Facility for Advanced Research Computing at Case Western Reserve University.

## Author contributions

M.E. developed the approach and designed the project. C.G., G.S., and M.H. supervised the project. N.H. and A.R.L. performed color analysis. A.R.L. performed color gamut optimization. A.R.L. and S.A.J. fabricated the samples. A.R.L., T.L., M.E., performed measurements. M.E. wrote the manuscript with inputs from N.H. and A.R.L. All authors discussed the results.

## Competing interests

A patent application has been filed on the Fano resonance optical coating scheme shown in this work. Patent #: US20220308264A1. Applicant: Case Western Reserve University, University of Rochester. Inventors: M.E., C.G., M.H., G.S. Status: pending.
