## [Peer Review File · Nature Communications]

Fano Resonant Optical coatings platform for Full Gamut and High Purity Structural ColorsREVIEWER COMMENTS

Reviewer #1 (Remarks to the Author):

This manuscript reports structural coloration based on Fano resonant optical coatings (FROCs), which were first demonstrated by the same authors in 2021 ("Fano-resonant ultrathin film optical coatings," Nat. Nanotechnol. 16, 440–446 (2021)), aiming at achieving a wide range of color gamut. The FROC consists of two coupled oscillators with distinct damping rates, thus providing a versatile optical platform to simulate Fano resonance. Moreover, it provides unusual optical properties that have not been accessed via traditional optical coatings; for example, a FROC-based beam splitter reflects and transmits the same color. However, in terms of structural coloration, the optimization process of the proposed structure is not systematic or well-organized. In other words, advanced technique or fine engineering for extending the color gamut is rarely presented. More importantly, there seems to be a lack of novel strategies or concepts beyond what they reported in the previous FROC studies. So, I believe that this manuscript is not radically new enough to be worthy of publication. The reviewer provides below some comments that will help the authors improve the manuscript for a future submission.

1. Fig. 2e and Fig. S3 are the same.
2. What was the reason for obtaining the spectra for the p-polarization in Fig. 1c?
3. The counterpart MDM film is not optimized. Please compare the author's structure to the best-performing tri-layer configuration in terms of color gamut.
4. Why did the authors choose SiO₂ as a capping layer? Please specify the corresponding design rule.

Reviewer #2 (Remarks to the Author):

Note: a pdf of these review comments is also attached.

Review of "Fano Resonant Optical coatings platform for Full Gamut and High Purity Structural Colors" submitted to Nature Communications.

Reviewer comments:

The core novelty of this work is the development of structural coloration based on the authors recently introduced Fano Resonant Optical Coating (FROC) concept [Ref. 16]. Compared to Ref. 16, this work makes a substantial advancement in exploring and demonstrating FROCs for structural color. In this work the FROC concept is well contrasted with Fabry-Perot (FP) coatings, for example where it reads:

“While Fabry-Perot nanocavities produce colors through selective absorption, mainly reflecting all colors except for the specific cavity resonance wavelength. On the other hand, FROCs produce colors through selective reflection (Fig. 1a).”

The authors demonstrate their devices through systematic and thorough numerical and experimental studies. Given the quality of their results, novelty of the work, and strong community interest in the topic at hand, this reviewer is inclined to favor publication in Nature Communications provided minor revisions and concerns can be adequately addressed. These items are noted below:

1.) The FROC concept is presented as a means to obtain ‘selective reflection’ maxima (“peaks”) rather than ‘selective absorption’ or reflectance minima (“dips”) typically associated with FP type structures. However, the authors should be careful to note that FP structures can in general provide either reflectance peaks or dips (not just dips). It may be worth reiterating to the readership why FP reflectance peaks are typically not preferred over FP reflectance dips based on ultra-thin films incorporating loss. A classic example of a reflectance peaking FP structure is SiO₂ on Si or SiN on Si for example. Presumably the work at hand outperforms those basic examples. The advantage should be made clear.

2.) The authors discuss color angle dependence in paragraph 3: “An ideal structural coloring platform should span a wide color gamut [...] and allowing control over the colors’ angle dependence”. Angular dependence is further examined in greater detail in Figure 5 where the authors state “For high refractive indices, FROCs color over nearly the entire range of incident angles”.

a. However, it is not clear if Fig. 5a is physically very meaningful, since large refractive indices would also be associated with a change in both n and k vs. just n .

b. Moreover, is the refractive index value cited in Fig. 5a applied to the entire spectrum or specified at a certain wavelength?

c. Exactly how practical material dispersion and loss would affect the results is somewhat unclear.

d. Perhaps there is a better way to demonstrate or show the angular dependence.

3.) The authors allude to anti-counterfeit measures as one prospective application several times, as well as the role of angle-dependence in such an application. However no relevant citation or literature example is provided.

4.) Figure 1a font sizes and graphics are relatively small and moderate quality. Recommend improving this figure.

5.) The authors should cite additional related works in this area. In particular the FROC concept could be interpreted as a combination of lossy dielectric coatings plus metal insulator metal FP structures. Two primary examples include these two:

- Kats, M., Blanchard, R., Genevet, P. et al. Nanometre optical coatings based on strong interference effects in highly absorbing media. *Nature Mater* 12, 20–24 (2013). <https://doi.org/10.1038/nmat3443>
- Zhongyang Li, Serkan Butun, and Koray Aydin, Large-Area, Lithography-Free Super Absorbers and Color Filters at Visible Frequencies Using Ultrathin Metallic Films, *ACS Photonics* 2015 2 (2), 183-188 DOI: 10.1021/ph500410u

Other comments/formatting items:

- i. The authors should include an illustration describing their experimental setup for measuring the reflection spectrum, at least in the supplementary information file.
- ii. Fig. 3c needs a scale bar.
- iii. A macroscopic image or photograph of an example experimental chip (e.g. used in Fig. 3a) would be a good addition to the supplementary information.
- iv. Decapitalize “Nanophotonic” (e.g. in abstract)
- v. Introduction, paragraph 1: A hyphen is not the correct way to start a list. It should be a colon.
- vi. Introduction, paragraph 2 the sentence "An ideal coloring platform..." is too long for a single sentence. Suggest breaking into two sentences.
- vii. Introduction, paragraph 3: Missing space after bold text for Figure 1a, "ofFROCs" rather than "of FROCs".
- viii. Results and Discussion, paragraph 1: Incomplete sentence: "While Fabry-Perot nanocavities produce colors through selective absorption, mainly reflecting all colors except for the specific cavity resonance wavelength."
- ix. Results and Discussion, paragraph 2: "FROCs access the huge color gamut..." Use of "the huge color gamut" should be replaced by something like "FROCs access a wide color gamut..."
- x. Results and Discussion, paragraph 3: The abbreviation "MDM" is not defined. Metal-dielectric-metal?
- xi. Results and Discussion, paragraph 3-4: Rough paragraph transition. One paragraph ends with "High purity green colors were difficult to obtain using conventional FROCs and were obtained using silica

capped FROCs (red stars) as we will discuss below." while the next begins with "High purity green and red colors are difficult to obtain." The redundancy should be avoided.

xii. Results and Discussion, paragraph 4: First two sentences writing is unclear and can be improved.

xiii. Conclusion and outlook: "nanostructures based structural coloring" could be written better as "nanostructure-based structural coloring" or similar.

Fano Resonant Optical coatings platform for Full Gamut and High Purity Structural Colors

Mohamed ElKabbash^{1,2, †, *}, Nathaniel Hoffman^{3, †}, Andrew R. Lininger^{3, †}, Sohail A. Jalil¹, Theodore Letsou³, Michael Hinczewski^{3, *}, Giuseppe Strangi^{3,4, *}, and Chunlei Guo^{1, *}

1. The Institute of Optics, University of Rochester, Rochester, NY 14627, USA.
2. Current address: Research Laboratory of Electronics, MIT, Cambridge, MA, 02139, USA
3. Department of Physics, Case Western Reserve University, 10600 Euclid Avenue, Cleveland, Ohio 44106, USA.
4. CNR-NANOTEC and the Department of Physics University of Calabria, Rende (Italy).

* Corresponding emails: melkabba@mit.edu (M.E.), mxh605@case.edu (M.H.), gxs284@case.edu (G.S.), guo@optics.rochester.edu (C.G.).

†These authors contributed equally.

Summary of major revisions

We have made extensive revisions to the manuscript including:

- An extensive ‘apples-to-apples’ comparison to the best performing thin-film structural coloring platforms, highlighting the extraordinary performance of FROCs
- A detailed investigation of the tunable iridescence properties of structural color FROCs with varying material composition
- Globalized optimization of structural color FROCs to maximize gamut coverage and purity.
- 4 updated figures and 9 new supplementary figures
- 9 additional citations in the manuscript

Specifically in response to Reviewer #1:

We have made the following major revisions, as well as highlighting the novelty of investigating FROCs for structural coloring applications:

- Including a detailed description of the optimization process to maximize purity
- Comparing structural color FROCs to the best performing Fabry-Perot systems, and the best performing thin-film structural color platforms in the literature
- Providing justification for the SiO₂ capping layer
- Investigating the polarization sensitivity of the coloring response

Specifically in response to Reviewer #2:

We have made the following major revisions:

- Comparing structural coloring FROCs with reflective peak Fabry-Perot systems
- Providing an in-depth analysis of the tunable iridescence properties of FROCs with 1) real deposition materials, 2) typical dispersive refractive indices, and 3) including extinction.
- Providing an illustration of the experimental setup

The point-by-point response begins on the following page.

Reviewer #1:

This manuscript reports structural coloration based on Fano resonant optical coatings (FROCs), which were first demonstrated by the same authors in 2021 ("Fano-resonant ultrathin film optical coatings," Nat. Nanotechnol. 16, 440–446 (2021)), aiming at achieving a wide range of color gamut. The FROC consists of two coupled oscillators with distinct damping rates, thus providing a versatile optical platform to simulate Fano resonance. Moreover, it provides unusual optical properties that have not been accessed via traditional optical coatings; for example, a FROC-based beam splitter reflects and transmits the same color. However, in terms of structural coloration, the optimization process of the proposed structure is not systematic or well-organized. In other words, advanced technique or fine engineering for extending the color gamut is rarely presented. More importantly, there seems to be a lack of novel strategies or concepts beyond what they reported in the previous FROC studies.

So, I believe that this manuscript is not radically new enough to be worthy of publication.

We thank the reviewer for their time and effort reviewing our manuscript and for their feedback. The authors agree with the reviewer that the optimization process could be better explained. In fact, following the reviewer's suggestion, we used our optimization method to maximize the color gamut coverage of FROCs and in the revised manuscript, **we show that the CIE gamut coverage of FROCs is unprecedented for thin-film platforms and even comparable with the state-of-the-art nanostructure based structural coloring approaches[1]**. Below we first detail the optimization process we adopted in this revised work. With this revision, we believe that the novelty of this work has become far more evident. We will provide a detailed discussion on the novelty in the proposed structural coloring approach based on FROCs.

Optimization process for structural color FROCs:

The simplicity of the FROC structural coloring platform leads to a relatively straightforward optimization problem. We adopted a least-squares optimization process (Trust Region Reflective algorithm) in the SciPy python package to simultaneously fit all layer thicknesses. Each final structure minimizes the root mean squared error (RMSE) x-y space distance to a maximum purity point on the outer ring of the gamut. A bounded 60-point random global fit was completed for each structure, performing the least-squares optimization at each instance with random initial layer thicknesses proposed from a design space of reasonable layer thicknesses. **Figure R1** shows the optimization process for sample blue, orange, and green colors. In all the cases shown, the RMSE x-y distance space is simple and has a smooth global minimum. Furthermore, the reflectance peak location depends mainly on the dielectric cavity thickness, which determines the Fabry-Perot cavity resonance wavelength (**Figure 1b**). Accordingly, the RMSE space for FROCs is simple and enjoys low dimensional correlation, meaning that each layer has an independently discernable effect on the optical response. These factors justify the adopted straightforward optimization protocol. In the case of the silica capping layer, the silica cap thickness was determined such that it simultaneously maximizes the reflectance and purity. Based on the considerations above, we believe this protocol is sufficient to fully cover the design space and that more sophisticated optimization techniques are unnecessary for this problem. The optimization code will be made freely available on GitHub (https://github.com/hincz-lab/structural_color_FROCs). We have also

added this information and figure in the supplementary material. We have revised the manuscript and added the following sentences (lines 189-209):

Finally, we assess the color range covered by silica capped FROCs. FROCs access a significantly larger color gamut since the wavelength of the selective reflection is tuned by simply changing the dielectric thickness. **Figure 5a** shows the CIE 1931 chromaticity coordinates of FROCs vs. FP cavities. The CIE gamut coverage for both approaches is maximized by employing a least-squares based optimization procedure to optimize the individual layer thicknesses to produce maximal purity (see Supplementary information, **Fig. S9**, and related text)²⁵. FP cavities are considered in two varieties: tuning the thickness of all three layers in the metal-dielectric-metal stack (all layers) or assuming the bottom metal layer is maximally reflective and opaque (opaque base). Although opaque base FP cavities are the most direct comparison with FROCs, a larger color space can be accessed by optimizing all three layers. At normal incidence FROCs cover 43.9% of the CIE gamut, while SiO₂-FROCs cover 61.3%. This includes a near complete (> 99%) coverage of the sRGB and Adobe RGB color spaces with a total covered area reaching 131.6% and 177.5% compared to the Adobe RGB and sRGB color subspaces (see Supplementary information, **Fig. S10**). On the other hand, FP cavities only cover 31.8% of the CIE color space even when optimizing all three layers. FROCs are also compared with several of the best performing thin-film structural coloring platforms in the recent literature (**Figure 5a**). The color response of these platforms are simulated based on the structures reported in the literature, but not further optimized. To the best of our knowledge, the CIE gamut coverage of FROCs is larger than any other thin-film structural coloring platform^{19,20,26,27}. In fact, optimized SiO₂-FROCs can cover a larger color gamut compared to the state-of-the-art 2D nanostructure-based structural coloring approaches.^{7,28}

In addition, Figure R1 is now added to the revised SI as figure S9.

Figure R1 Illustration of the optimization protocol for structural coloring using FROCs. **(a)** Three examples are presented for FROC structural coloring for blue, orange, and green, from top to bottom to optimize maximum purity levels. The RMSE x-y distance space is shown for different values of Ge (y), Ag (x), and TiO₂ (plot) layer thicknesses. In each case the RMSE space is relatively uncomplicated with a visible global minimum. **(b)** The Fabry-Perot resonance wavelength changes linearly with TiO₂ layer (cavity) thickness. The resonant wavelength for the FROC can be tuned by controlling the TiO₂ layer thickness. **(c)** CIE color space showing the target color response and optimized FROC system for the three cases presented in (a). The optimization protocol accurately predicts structures up to the maximum purity level (black line).

I. Novel claims presented in the current manuscript:

This work presents, for the first time, the development and systematic investigation of a structural coloring platform based on FROCs, as aspect which has not been previously explored. We find that FROCs out-perform other thin film coloring platforms in almost every metric. Accordingly, we are confident that this work will be of major impact for the structural coloring research community. Our study on the structural coloring properties of the Fano resonant optical coating presented in this manuscript is sufficiently novel to warrant its publication for the following reasons:

1- **The novelty of the structural coloring approach:** FROCs represent an innovative approach to thin-film structural coloring. Thin film interference based structural coloring (e.g., multilayer films[6], thin film nanocavities[7]), relies on either selective absorption or selective transmission, i.e., it is based on subtractive coloring. The subtractive coloring approach is very limiting when it comes to structural coloring particularly using thin films, because unlike nanostructures, it is not possible to perform color mixing to access the entire color gamut. Ghobadi, et. al.[1], attempted to address this problem through a multilayer design that controllably creates a selective reflection peak through selective absorption of two bands in the visible spectrum. However, the demonstrated colors lack purity and brightness. We add below **Figure 6e** in Ghobadi, et.al., to show the obtained red, green, and blue based on their approach. Note that to our knowledge this is the second best performing structural coloring platform compared to FROCs. On the other hand, FROCs allow for narrowband selective reflection since the reflection peak is determined by the metal-dielectric-metal nanocavity's resonance. **Fig. R2** shows the experimentally obtained red, green and blue based on the silica-capped FROC approach presented here. The strikingly pure and bright colors obtained testify for the impact of our structural coloring approach. **Figure R2** is now added to the supplementary information as **Fig. S5** and is also a part of **figure 3c** in the revised manuscript.

Figure 6e from Ghobadi, et. al.[1], selective reflectance structural coloring approach.

Figure R2| Photographs of experimentally fabricated silica capped FROC samples illustrating blue (top), green (middle), and red (bottom) structural color resonances. The samples are illuminated with a F11 type illuminant spectrum.

- 2- **Further boosting the purity through an additional thin film:** In the manuscript, we address the low purity of conventional FROCs when targeting green and red colors. We propose a new architecture by capping FROCs with a silica layer and optimize the layer thickness to boost the purity of green and red colors. The results are shown in **Figure R3** and **Fig. R4** where silica capped FROCs significantly widen the accessible color gamut (which is equivalent to accessing higher color purity).
- 3- **Superior performance compared to existing thin-film based structural coloring platforms:** To prove our claim that FROC outperform all other platforms in the recent literature, **Fig. R3** and **Fig. R4** compare FROCs with optimized Fabry-Perot cavities by optimizing the three layers, or assuming the bottom layer is opaque and optimizing the top two layers.

Figure R3 | CIE 1931 color space showing the colors corresponding to the calculated reflection spectrum of optimized FROCs, Silica Capped FROC, Fabry Perot cavities either by optimizing all three layers of the Fabry Perot cavities or optimizing the top two layers and assuming the bottom layer to be optically opaque.

Figure R4 | Gamut coverage for each of the structural coloring platforms shown in Figure R3, compared with FROCs. Silica capped FROCs produce the greatest gamut coverage over a wide angular range.

We also added a new figure, **Figure 5a** in the revised manuscript, to compare FROCs with Fabry Perot cavities and existing thin film structure coloring approaches. The SiO₂-FROC approach covers 61.2% of the CIE color space **making it the best performing structural coloring approach to the extent of our knowledge, even comparable to 2D nanostructure-based structural coloring systems**[4, 5].

Figure 5| Accessible color gamut, brightness and overall performance of FROCs: (a) CIE 1931 chromaticity diagram of FROCs and Silica-capped FROCs vs. Fabry-Perot cavities and other reported thin-film based coloring approaches from literature at normal incidence. **(b)** Measured reflectance from different FROCs showing a peak reflectance ranging from 0.63-0.85. The high reflectance corresponds to high brightness, a desirable property in structural colors. **(c)** Comparison between the coloring performance of FROC vs. other structural coloring platforms.

- 4- **Studying the angular dependence of the produced colors:** Another important point of this work is a comprehensive study of the ability to engineer the angular dependence of the produced colors. From an optimization and iridescence control standpoint, FROCs are an incredibly advantageous system since the thickness and material properties of a single layer (the dielectric cavity) can independently tune the resonance wavelength and iridescence properties while other features (for example purity) are largely unchanged. This property is not as easily obtained in many other structural coloring platforms, since the typically higher degree of parameter coupling in the optical response can lead to an altered color resonance when controlling the iridescence properties. The angular dependence is elaborately discussed in Figure 4 and related text the revised manuscript.

The reviewer provides below some comments that will help the authors improve the manuscript for a future submission.

1. Fig. 2e and Fig. S3 are the same.

We thank the reviewer for pointing the mistake out. We removed the redundant figure in the SI.

2. What was the reason for obtaining the spectra for the p-polarization in Fig. 1c?

Figure 1b and **Figure 1c** are meant to show how a Fabry-Perot cavity that selectively absorbs a certain color is converted into a FROC that selectively reflects almost the same wavelength range simply by adding a 15nm Ge layer. We performed the measurements using an ellipsometer where light is polarized. We revised the paragraph in the manuscript that explains **Fig. 1b** and **Fig. 1c** to avoid any confusion. The revised paragraph (lines: 78-85) now reads:

“**Figure 1b** shows the measured *p*-polarized reflection spectrum of Fabry-Perot nanocavities consisting of Ag (20 nm)-TiO₂-Ag (100 nm) by varying the thickness of the TiO₂ film from 35 nm to 70 nm (see Supplementary information, Fig. S1). **Figure 1c** shows the measured reflection spectrum of the same Fabry-Perot nanocavities after adding a 15 nm Ge layer to convert them into FROCs. The reflection band produced by the FROCs correspond to the absorption band produced by the FP cavity. Consequently, by changing the dielectric thickness/refractive index of the FP cavity, FROCs can reflect a relatively narrow range of wavelengths that can be tuned across the visible spectrum.”

Furthermore, we have compared the colorimetric properties of FROCs illuminated by p-polarized and un-polarized light. A sample of the reflected colors for cavity thicknesses of 60 nm, 70 nm, and 80 nm, respectively, is shown below in **Fig. R5**. The p-polarized reflection (top) is compared against the un-polarized reflection (bottom) as a function of angle. At normal incidence the two responses are indistinguishable, and the color resonance is not notably different until reaching an angle of incidence near 50 deg. From these results we can conclude that the central results in this work are unchanged regardless of the polarization state.

Figure R5 Comparison of the reflected color from FROCs as a function of incident angle for dielectric cavity thicknesses 60 nm (left), 70 nm (center) and 80 nm (right). The color for p-polarized illumination (top) is compared with the un-polarized reflectance (bottom).

3. The counterpart MDM film is not optimized. Please compare the author’s structure to the best performing tri-layer configuration in terms of color gamut.

We thank the reviewer for their suggestion. We adopted the least-squares optimization method we used for FROCs and applied it to Fabry Perot (MDM) cavities. **Figure R3** and **Fig.**

R4 above compare the CIE color space coverage of FROCs vs. Fabry-Perot cavities. This includes a full optimization of all layer thicknesses in the Fabry-Perot cavities (All Layers). The results are clear; FROCs provide a much large color gamut coverage as well as color purity compared to Fabry-Perot cavities. At normal incidence, FROCs cover 48.9% of the entire CIE gamut, while SiO₂-FROCs cover 61.2%. At higher incidence angles this increases to 62% coverage for both FROCs and SiO₂-FROCs. On the other hand, opaque base Fabry-Perot cavities only cover 4% of the CIE gamut, which increases to 31.8% when all three layers are thickness-optimized. To the best of our knowledge, the gamut coverage of FROCs is also larger than any other thin-film structural coloring approach in the current literature [2-5].

4. Why did the authors choose SiO₂ as a capping layer? Please specify the corresponding design rule.

We thank the reviewer for requesting this clarification. As we detailed in the manuscript, due to the additional reflection peaks that appear when targeting colors longer wavelengths, we added a dielectric capping layer as an impedance matching layer to suppress the reflectance at shorter wavelengths (**Fig. 2g** and **Fig. 2h**). Any lossless dielectric would have worked in this capacity.

We added the following sentences to the revised manuscript to clarify this point further:

“We note that any lossless dielectric other than SiO₂ can act as a capping layer which is meant to act as an impedance matching layer that further suppresses the reflection from the broadband absorber (**Fig. 2g** and **Fig. 2h**).”

Reviewer #2:

The core novelty of this work is the development of structural coloration based on the authors recently introduced Fano Resonant Optical Coating (FROC) concept [Ref. 16]. Compared to Ref. 16, this work makes a substantial advancement in exploring and demonstrating FROCs for structural color. In this work the FROC concept is well contrasted with Fabry-Perot (FP) coatings, for example where it reads:

“While Fabry-Perot nanocavities produce colors through selective absorption, mainly reflecting all colors except for the specific cavity resonance wavelength. On the other hand, FROCs produce colors through selective reflection (Fig. 1a).”

The authors demonstrate their devices through systematic and thorough numerical and experimental studies. Given the quality of their results, novelty of the work, and strong community interest in the topic at hand, this reviewer is inclined to favor publication in Nature Communications provided minor revisions and concerns can be adequately addressed. These items are noted below:

The authors greatly appreciate the reviewer’s effort and time reviewing our manuscript and welcome the reviewer’s constructive feedback.

1.) The FROC concept is presented as a means to obtain ‘selective reflection’ maxima (“peaks”) rather than ‘selective absorption’ or reflectance minima (“dips”) typically associated with FP type structures. However, the authors should be careful to note that FP structures can in general provide either reflectance peaks or dips (not just dips). It may be worth reiterating to the readership why FP reflectance peaks are typically not preferred over FP reflectance dips based on ultra-thin films incorporating loss. A classic example of a reflectance peaking FP structure is SiO₂ on Si or SiN on Si for example. Presumably the work at hand outperforms those basic examples. The advantage should be made clear.

We agree with the reviewer that Fabry-Perot type structures can be tuned to experience a reflectance peak. We searched the literature for works that attempted realizing structural coloring based on selective reflection. Ghobadi, et. al.[1], attempted to address this problem through a multilayer design that controllably creates a selective reflection peak through selective absorption of two bands in the visible spectrum. However, the demonstrated colors lack purity and brightness. We add below **Figure 6e** in Ghobadi, et.al., to show the obtained red, green, and blue based on their approach. Note that this is, to the best of our knowledge, the second best performing thin-film structural coloring platform compared to FROCs. On the other hand, FROCs allow for narrowband selective reflection since the reflection peak is determined by the metal-dielectric-metal nanocavity’s resonance. **Fig. R2** shows the experimentally obtained red, green and blue

based on the silica-capped FROC approach presented here. The strikingly pure and bright colors obtained testify for the impact of our structural coloring approach. **Figure R2** is now added to the supplementary information as **Fig. S5** and portions of the samples are added in **Figure 3c**. Note that **Figure 3a** and **Fig. 3b** in the revised manuscript are also experimentally obtained, however, it is not aesthetically pleasing to add arrays of glass slides with the thin film coating on top so we opted to only present a cropped part of each slide to show the colors obtained by varying the thickness.

Figure 6e from Ghobadi, et. al.[1], selective reflectance structural coloring approach.

Figure R1 Photographs of experimentally fabricated silica capped FROC samples illustrating blue (top), green (middle), and red (bottom) structural color resonances. The samples are illuminated with a F11 type illuminant spectrum.

In fact, through further optimizing our FROCs using least-squares optimization **we show that the CIE gamut coverage of FROCs is unprecedented for thin-film platforms and even comparable with the state-of-the-art nanostructure based structural coloring approaches[1].** We added a comparison between the color gamut coverage of our approach and several existing thin films based structural coloring approaches in the revised **Figure 5** shown below,

Figure 5| Accessible color gamut, brightness and overall performance of FROCs: (a) CIE 1931 chromaticity diagram of FROCs and Silica-capped FROCs vs. Fabry-Perot cavities and other reported thin-film based coloring approaches from literature at normal incidence. **(b)** Measured reflectance from different FROCs showing a peak reflectance ranging from 0.63-0.85. The high reflectance corresponds to high brightness, a desirable property in structural colors. **(c)** Comparison between the coloring performance of FROC vs. other structural coloring platforms.

We added the following sentences to further highlight the novelty of our approach (lines 189-209):

“

Finally, we assess the color range covered by silica capped FROCs. FROCs access a significantly larger color gamut since the wavelength of the selective reflection is tuned by simply changing the dielectric thickness. **Figure 5a** shows the CIE 1931 chromaticity coordinates of FROCs vs. FP cavities. The CIE gamut coverage for both approaches is maximized by employing a least-squares based optimization procedure to optimize the individual layer thicknesses to produce maximal purity (see Supplementary information, **Fig. S9**, and related text)²⁵. FP cavities are considered in two varieties: tuning the thickness of all three layers in the metal-dielectric-metal stack (all layers) or assuming the bottom metal layer is maximally reflective and opaque (opaque base). Although opaque base FP cavities are the most direct comparison with FROCs, a larger color space can be accessed by optimizing all three layers. At normal incidence FROCs cover 43.9% of the CIE gamut, while SiO₂-FROCs cover 61.3%. This includes a near complete (> 99%) coverage of the sRGB and Adobe RGB color spaces with a total covered area reaching 131.6% and 177.5% compared to the Adobe RGB and sRGB color subspaces (see Supplementary information, **Fig. S10**). On the other hand, FP cavities only cover 31.8% of the CIE color space even when optimizing all three layers. FROCs are also compared with several of the best performing thin-film structural coloring platforms in the recent literature (**Figure 5a**). The color response of these platforms are simulated based on the structures reported in the literature, but not further optimized. To the best of our knowledge, the CIE gamut coverage of FROCs is larger than any other thin-film structural coloring platform^{19,20,26,27}. In fact, optimized SiO₂-FROCs can cover a larger color gamut compared to the state-of-the-art 2D nanostructure-based structural coloring approaches.^{7,28}

We also consider the system suggested by the reviewer. Adding a lossless dielectric on Si . e.g., SiN, results in reflection peaks. However, as shown below **Figure R2**, the peaks maximum reflectance is < 0.5 which means the colors suffer from low brightness. In addition, the peaks are broad, and multiple peaks appear as the cavity thickness increases. This means that the colors lack purity. To further verify our claim regarding purity, we plot the observed colors on the CIE color space. As seen in figure R3, the FROC produce a high purity color (far from the white point), while the SiN/Si system produce low purity colors for the SiN thicknesses used in the calculation done in **Fig. R2**. We added Figure R2 and Fig. R3 in the Supplementary Information as **Figure S3 and Fig. S4**.

Figure R2| Comparison of the reflectance for SiN on a Si substrate vs. FROCs. The SiN/Si system is simulated for a range of thicknesses. Although the Fabry-Perot resonance results in a reflection peak, the colors suffer from low brightness.

Figure R3| CIE color space representation for the reflectance spectra presented in Figure R2. The color purity of the FROC is significantly larger than that of SiN/Si system.

2.) The authors discuss color angle dependence in paragraph 3: “An ideal structural coloring platform should span a wide color gamut [...] and allowing control over the colors’ angle dependence”. Angular dependence is further examined in greater detail in Figure 5 where the authors state “For high refractive indices, FROCs color over nearly the entire range of incident angles”.

a. However, it is not clear if Fig. 5a is physically very meaningful, since large refractive indices would also be associated with a change in both n and k vs. just n .

Figure 5a in the original manuscript was intended to demonstrate the tunable iridescence properties of the FROC platform, and that they can be controlled by manipulating the refractive index of the material in the cavity. We agree with the reviewer that the data in **Fig. 5a** could be more meaningful if the refractive indices corresponded with real materials (realistic dispersions and including extinction). However, it should be noted that the FROC platform is designed to utilize a lossless or low-loss dielectric as the cavity material. To make **Fig. 5a** realistic, we now only consider refractive indices up to 2.4 which is the case for several lossless dielectrics, e.g., diamond and Silicon Carbide. See the revised figure, **Fig. 4a** in the revised manuscript.

Finally, we believe that utilizing a dispersion-less material in this figure is less confusing for the reader since the full dispersion adds another layer of complexity to the figure. We consider dispersion and losses in our reply to the point (c.) below.

b. Moreover, is the refractive index value cited in Fig. 5a applied to the entire spectrum or specified at a certain wavelength?

The refractive index in **Fig. 5a** is applied to the entire spectrum. Each color in the swatch array is calculated from the entire spectrum corresponding to the parameters quoted (dispersion-less refractive index and angle). A dispersion-less refractive index was chosen for simplicity since dispersive refractive indices requires knowledge over the entire visible region and adds an additional layer of complexity.

c. Exactly how practical material dispersion and loss would affect the results is somewhat unclear.

We agree with the reviewer that it is important to investigate practical dispersions for the cavity material. We have followed two pathways to investigate the iridescence properties of FROCs with practical material dispersion:

1 – We use the Cauchy dispersion model for the refractive index with a constant extinction. This model is a good approximation for many lossless and low-loss dielectric materials in the visible range. The wavelength dependent refractive index is given by:

$$n = A + \frac{B}{\lambda^2}$$

In **Figure R5** below, we consider Cauchy dispersions with A in the range of 1.0 to 4.0, B of 0.025, and k in the range of $1E-3$ to $1E-1$ (see **Figure R6** for the corresponding dispersions). The results are qualitatively similar to those presented in Fig. 5a in the original manuscript, with the exception of a diminished structural coloring purity with increasing extinction which is likely due to the strong impedance mismatch (increased overall reflectance). Note that with FROCs we are generally considering cavity materials with low loss ($k \ll 0.1$). We have included these results in the supplementary material as **Figures S7** and **Figure S8**.

Figure R5] Iridescence properties of FROCs utilizing cavity materials with a Cauchy dispersion and constant extinction of 1E-3 (left), 1E-2 (center), and 1E-1 (right). The results are qualitatively similar to the simulated iridescence with dispersionless cavity materials.

Figure R6] Cauchy dispersions considered in **Figure R5**, for various values of a , with $b=0.025$. These dispersions represent a good approximation of many lossless dielectric materials in the visible range.

We also added a sentence in the main manuscript to point the readers to the relevant SI section, lines (172-175):

“However, introducing typical losses to the cavity material does not affect the general behavior captured in **Fig. 4a** (Supplementary information, Fig. S7 and Fig. S8). “

2 – To assess the iridescence of realistic materials, we consider FROCs fabricated with different dispersive dielectrics. In **Figure R7**, we consider amorphous Si (a-Si), TiO₂, SiO₂, and LiF as cavity materials with dispersive refractive indices spanning the range of 1.4 - 3.5. We vary

the cavity's thickness and the incidence angle. The refractive indices were obtained from refractiveindex.info. These results show that iridescence can be controlled by choosing the appropriate dielectric. In addition, by adding a high refractive index material, FROCs can provide minimal changes in the structural color over a wide range (0 - 70 deg) of angles.

Figure R7 Iridescence properties of FROCs with a range of cavity materials at different refractive indices. **(a)** Iridescence plots of FROCs with different cavity materials (a-Si, TiO₂, SiO₂, and LiF) and with a range of cavity thicknesses. The iridescence properties can be controlled by changing the material in the cavity. For high index materials, the color is relatively consistent for a wide range of incidence angles. **(b)** Gamut showing the color changes for TiO₂ (squares) and LiF (stars) cavity FROCs varying the angle of incidence from 0 deg to 65 deg at several cavity thicknesses. **(c)** Real refractive indices for the cavity materials in **(c)**.

In the revised manuscript, we added a separate figure, **Figure 4** in the revised manuscript, dedicated to the iridescence properties of FROCs. The figure is shown below for convenience. **Figure R7** is added to the supplementary information for completeness as **Fig. S6**.

Figure 4| Angle dependence of FROCs: (a) Swatch array of a silica capped FROC for a constant optical length by varying the refractive index and incidence angle. The angle dependence can be controlled by the refractive index of the dielectric cavity. (b) and (c) show a swatch array as a function of the cavity thickness and incidence angle for a FROC with real dispersive materials TiO_2 ($n \sim 2.3$) and LiF ($n \sim 1.4$) as the dielectric cavity material.

d. Perhaps there is a better way to demonstrate or show the angular dependence.

We appreciate the reviewer's suggestion. In literature, we have found that the angle dependence is typically assessed experimentally by taking pictures at different angles and showing qualitative similarity. Our approach, we believe, is more rigorous since we calculate the spectrum for each angle from a given design, convert that to the corresponding perceived color and then add it to the swatch array. Furthermore, in the supplementary materials we provide a gamut showing the color changes for various thicknesses of TiO_2 cavity FROCs over a range of incidence angles. This eliminates any discrepancy due to human color perception. We are happy to examine alternative suggestions for presenting this information.

Mathematically, the angle dependence of FROC's resonant reflection mode depends entirely on the properties of the MDM cavity. The reflection peak wavelength λ_{max} , dependence on the incident angle is thus given by:

$$\frac{1}{\lambda_{max}(\theta)} \frac{d\lambda_{max}(\theta)}{d\theta} H(\lambda_{max}(\theta), \theta, n_d) \frac{\cos\theta \sin\theta}{n_d^2 - \theta^2},$$

where $H(\lambda_{max}(\theta), \theta, n_d)$ is a dimensionless function that depends on solely on θ through λ_{max} . As n_d increases to values $\gg 1$, the above expression decreases as n_d^{-2} . Accordingly, the iridescence of FROCs can be mitigated significantly by using a high index dielectric. We demonstrate this fact in **Fig. 4** in the revised manuscript. We added the discussion above to the text associated with **Figure S6**.

3.) The authors allude to anti-counterfeit measures as one prospective application several times, as

well as the role of angle-dependence in such an application. However no relevant citation or literature example is provided.

- We thank the reviewer for reminding us. We added the following references that discuss the application of iridescent colors to anti-counterfeit measures[8, 9]. In fact, it is very common for credit cards to have iridescent colors on them since iridescent drawings are harder to imitate. The revised manuscript reads:

“Note that colors’ iridescence is a desired property in some applications, e.g., anti-counterfeit measures^{22,23}. “

4.) Figure 1a font sizes and graphics are relatively small and moderate quality. Recommend improving this figure.

We revised all the figures in the manuscript to make the font sizes easily readable.

5.) The authors should cite additional related works in this area. In particular the FROC concept could be interpreted as a combination of lossy dielectric coatings plus metal insulator metal FP structures. Two primary examples include these two:

- Kats, M., Blanchard, R., Genevet, P. et al. Nanometre optical coatings based on strong interference effects in highly absorbing media. *Nature Mater* 12, 20–24 (2013). <https://doi.org/10.1038/nmat3443>

- Zhongyang Li, Serkan Butun, and Koray Aydin, Large-Area, Lithography-Free Super Absorbers and Color Filters at Visible Frequencies Using Ultrathin Metallic Films, *ACS Photonics* 2015 2 (2), 183-188 DOI: 10.1021/ph500410u.

We really appreciate the reviewer’s suggestion for the two seminal works in the field. We referenced them heavily in the original *Nature Nanotechnology* work and we have now added them to the revised manuscript (references 17 and 18)

Other comments/formatting items:

We appreciate the effort of the reviewer in providing this comprehensive review.

i. The authors should include an illustration describing their experimental setup for measuring the reflection spectrum, at least in the supplementary information file.

A figure detailing the optical experimental setup has been added to the supplementary information, **Fig. S1**.

ii. Fig. 3c needs a scale bar.

We added the scale bar to the revised figure, now **Fig. 3d**, in the revised manuscript.

iii. A macroscopic image or photograph of an example experimental chip (e.g. used in Fig. 3a) would be a good addition to the supplementary information.

The colors shown in Fig. 3 a,b are all cropped photographic images of fabricated FROCs and MDMs, respectively, illuminated with a F11 illuminant. Additionally, we have added some photographic images of the full glass slides with SiO₂- FROCs in the supplementary materials, **Fig. S5**. See our reply to point 1 made by the reviewer.

iv. Decapitalize “Nanophotonic” (e.g. in abstract)

We fixed the mistake. Thank you.

v. Introduction, paragraph 1: A hyphennm;- is not the correct way to start a list. It should be a colon.

We corrected the error. Thank you.

vi. Introduction, paragraph 2 the sentence "An ideal coloring platform..." is too long for a single sentence. Suggest breaking into two sentences.

We agree with the reviewer. We modified the sentences and now it reads:

“An ideal structural coloring platform should satisfy several requirements. It should span a wide color gamut producing colors with high and controllable purity, i.e., how monochromatic or pure the color is, and brightness, i.e., the relative intensity of the reflected color. The platform should also allow controlling over iridescence, i.e., the angle dependence of observed colors which is particularly common for colors generated via interference. For many applications, structural coloring should be scalable and inexpensive to fabricate. However, no existing scheme has been shown to satisfy all the above qualities simultaneously[6-9].”

vii. Introduction, paragraph 3: Missing space after bold text for Figure 1a, "ofFROCs" rather than "of FROCs".

We agree that it looks as if there is no space. However, there is a space that, for some reason, is not appearing clearly. We hope that in the production, this problem is resolved.

viii. Results and Discussion, paragraph 1: Incomplete sentence: "While Fabry-Perot nanocavities produce colors through selective absorption, mainly reflecting all colors except for the specific cavity resonance wavelength."

We again thank the reviewer for a thorough and rigorous review. We revised the sentence and now it reads:

“Fabry-Perot nanocavities produce colors through selective absorption, mainly reflecting all colors except for the specific cavity resonance wavelength”

ix. Results and Discussion, paragraph 2: "FROCs access the huge color gamut..." Use of "the huge color gamut" should be replaced by something like "FROCs access a wide color gamut..."

We agree with the reviewer. The sentence now reads:

“FROCs access a significantly larger color gamut since they provide selective reflection at different wavelengths by simply changing the dielectric thickness”

x. Results and Discussion, paragraph 3: The abbreviation "MDM" is not defined. Metal-dielectric-metal?

The reviewer is correct. MDMs are the same as Fabry Perot cavities. For consistency, we removed the term MDM and replaced it with Fabry-Perot.

xi. Results and Discussion, paragraph 3-4: Rough paragraph transition. One paragraph ends with "High purity green colors were difficult to obtain using conventional FROCs and were obtained using silica capped FROCs (red stars) as we will discuss below." while the next begins with "High purity green and red colors are difficult to obtain." The redundancy should be avoided.

xii. Results and Discussion, paragraph 4: First two sentences writing is unclear and can be improved.

We revised this section in the manuscript significantly by merging the three systems under study in Figure 2. The new section now reads as follows (lines 114-129):

However, close inspection of **Figure 2c** and **Figure 2d** reveals that high purity green and red colors are still difficult to obtain. This is because pure red and green colors require having the Fano resonance peak at longer wavelengths. On the other hand, the measured reflection from the Ge-Ag broadband absorber is > 0.15 at short wavelengths < 500 nm (**Fig. 2g**). Consequently, the color purity drops since shorter wavelengths are strongly reflected. To suppress the high reflection at shorter wavelengths, we add a 50nm silicon dioxide (SiO_2) capping layer **Figure 2g**. **Figure 2h** shows the measured reflection for a FROC that produces a green color with and without the SiO_2 capping layer. The SiO_2 -FROCs exhibit significantly reduced reflection at short wavelengths and relatively small change near the resonance peak. This combination leads to an overall increase in the colorimetric purity. **Figure 2e** and **Figure 2f** show the swatch array and corresponding color purity of the SiO_2 -FROCs design, respectively. By comparing **Figures 2e** and **2f**, with **Figures 2c** and **2d**, respectively, SiO_2 -FROCs are shown to produce greater purities compared to the original FROC structure. A purity level $> 99\%$ can be reached (Supplementary information, **Fig. S2**) which is higher than other thin-film structural color platforms⁷. We note that any lossless dielectric other than SiO_2 can act as a capping layer which is meant to act as an impedance matching layer that further suppresses the reflection from the broadband absorber (see **Fig. 2g** and **Fig. 2h**).

xiii. Conclusion and outlook: "nanostructures based structural coloring" could be written better as "nanostructure-based structural coloring" or similar.

We revised the phrasing as suggested by the reviewer.

References:

- [1] A. Ghobadi, H. Hajian, M.C. Soydan, B. Butun, E. Ozbay, Lithography-free planar band-pass reflective color filter using a series connection of cavities, *Scientific Reports*, 9 (2019) 1-11.
- [2] A. Ghobadi, H. Hajian, M. Gokbayrak, B. Butun, E. Ozbay, Bismuth-based metamaterials: from narrowband reflective color filter to extremely broadband near perfect absorber, *Nanophotonics*, 8 (2019) 823-832.
- [3] J. Zhao, M. Qiu, X. Yu, X. Yang, W. Jin, D. Lei, Y. Yu, Defining Deep-Subwavelength-Resolution, Wide-Color-Gamut, and Large-Viewing-Angle Flexible Subtractive Colors with an Ultrathin Asymmetric Fabry–Perot Lossy Cavity, *Advanced Optical Materials*, 7 (2019) 1900646.
- [4] M. Song, L. Feng, P. Huo, M. Liu, C. Huang, F. Yan, Y.-q. Lu, T. Xu, Versatile full-colour nanopainting enabled by a pixelated plasmonic metasurface, *Nature Nanotechnology*, DOI (2022) 1-8.
- [5] B. Yang, W. Liu, Z. Li, H. Cheng, D.-Y. Choi, S. Chen, J. Tian, Ultrahighly saturated structural colors enhanced by multipolar-modulated metasurfaces, *Nano letters*, 19 (2019) 4221-4228.
- [6] Z. Xuan, J. Li, Q. Liu, F. Yi, S. Wang, W. Lu, Artificial structural colors and applications, *The Innovation*, 2 (2021) 100081.
- [7] Z. Li, S. Butun, K. Aydin, Large-area, lithography-free super absorbers and color filters at visible frequencies using ultrathin metallic films, *Acs Photonics*, 2 (2015) 183-188.
- [8] J.A. Dobrowolski, F.C. Ho, A. Waldorf, Research on thin film anticounterfeiting coatings at the National Research Council of Canada, *Appl. Opt.*, 28 (1989) 2702-2717.
- [9] M. Chang, H. Hu, H. Quan, H. Wei, Z. Xiong, J. Lu, P. Luo, Y. Liang, J. Ou, D. Chen, An iridescent film of porous anodic aluminum oxide with alternately electrodeposited Cu and SiO₂ nanoparticles, *Beilstein journal of nanotechnology*, 10 (2019) 735-745.

- [1] M. Song, L. Feng, P. Huo, M. Liu, C. Huang, F. Yan, Y.-q. Lu, T. Xu, Versatile full-colour nanopainting enabled by a pixelated plasmonic metasurface, *Nature Nanotechnology*, DOI (2022) 1-8.
- [2] A. Ghobadi, H. Hajian, M.C. Soydan, B. Butun, E. Ozbay, Lithography-free planar band-pass reflective color filter using a series connection of cavities, *Scientific Reports*, 9 (2019) 1-11.
- [3] A. Ghobadi, H. Hajian, M. Gokbayrak, B. Butun, E. Ozbay, Bismuth-based metamaterials: from narrowband reflective color filter to extremely broadband near perfect absorber, *Nanophotonics*, 8 (2019) 823-832.
- [4] J. Zhao, M. Qiu, X. Yu, X. Yang, W. Jin, D. Lei, Y. Yu, Defining Deep-Subwavelength-Resolution, Wide-Color-Gamut, and Large-Viewing-Angle Flexible Subtractive Colors with an Ultrathin Asymmetric Fabry–Perot Lossy Cavity, *Advanced Optical Materials*, 7 (2019) 1900646.
- [5] C.-S. Park, S.-S. Lee, Vivid coloration and broadband perfect absorption based on asymmetric Fabry–Pérot nanocavities incorporating platinum, *ACS Applied Nano Materials*, 4 (2021) 4216-4225.
- [6] A. Kristensen, J.K. Yang, S.I. Bozhevolnyi, S. Link, P. Nordlander, N.J. Halas, N.A. Mortensen, Plasmonic colour generation, *Nature Reviews Materials*, 2 (2016) 1-14.

- [7] Y. Shen, V. Rinnerbauer, I. Wang, V. Stelmakh, J.D. Joannopoulos, M. Soljacic, Structural colors from Fano resonances, *Acs Photonics*, 2 (2015) 27-32.
- [8] B. Yang, W. Liu, Z. Li, H. Cheng, D.-Y. Choi, S. Chen, J. Tian, Ultrahighly saturated structural colors enhanced by multipolar-modulated metasurfaces, *Nano letters*, 19 (2019) 4221-4228.
- [9] K. Kumar, H. Duan, R.S. Hegde, S.C. Koh, J.N. Wei, J.K. Yang, Printing colour at the optical diffraction limit, *Nature nanotechnology*, 7 (2012) 557-561.

REVIEWER COMMENTS

Reviewer #1 (Remarks to the Author):

I have comprehensively evaluated the manuscript entitled "Fano Resonant Optical coatings platform for Full Gamut and High Purity Structural Colors". I posit that the authors have, to a significant extent, addressed the concerns raised by both reviewers. One of the significant concerns was to provide ample justification for the superiority of the developed Fano resonant optical coatings (FROCs) over the current Fabry-Perot (FP) cavity-based multilayers, particularly in terms of color variations. The authors have employed an optimization algorithm to determine each optimized structure and quantitatively compared their results. Nonetheless, I still hold some reservations regarding the extent to which the optimized FROCs and FP-based multilayers surpass the benchmark studies, specifically (i) C. Ji et al. "High-color-purity, angle-invariant, and bidirectional structural colors based on higher-order resonances" published in Opt. Lett. 2019, and (ii) J. Lee et al. "High-purity reflective color filters based on thin film cavities embedded with an ultrathin Ge₂Sb₂Te₅ absorption layer" published in Nanoscale Adv. 2020. While the authors have made a quantitative comparison of each optimized structure through an optimization algorithm, a direct comparison with the aforementioned studies would provide a better understanding of the contributions of this work to the field. Therefore, I would appreciate it if the authors could include a discussion of how their results compare to those of the cited studies. This would provide additional insights into the novelty and significance of their proposed platform and its potential impact in the field of optical coatings.

Reviewer #2 (Remarks to the Author):

The authors have made substantial revisions to enhance the quality of their manuscript. All my prior concerns are now addressed and I recommend this work for timely publication.

Response To Decision Letter

Manuscript #: NCOMMS-22-38037

Title: Fano Resonant Optical coatings platform for Full Gamut and High Purity Structural Colors

Authors: M. ElKabbash et. al

Summary of major revisions

We have updated the manuscript to include a comparison between the structural coloring abilities of FROCs and the two additional benchmark studies indicated by the reviewer.

Responses in blue text

Reviewer #1:

I have comprehensively evaluated the manuscript entitled "Fano Resonant Optical coatings platform for Full Gamut and High Purity Structural Colors". I posit that the authors have, to a significant extent, addressed the concerns raised by both reviewers. One of the significant concerns was to provide ample justification for the superiority of the developed Fano resonant optical coatings (FROCs) over the current Fabry-Perot (FP) cavity-based multilayers, particularly in terms of color variations. The authors have employed an optimization algorithm to determine each optimized structure and quantitatively compared their results. Nonetheless, I still hold some reservations regarding the extent to which the optimized FROCs and FP-based multilayers surpass the benchmark studies, specifically (i) C. Ji et al. "High-color-purity, angle-invariant, and bidirectional structural colors based on higher-order resonances" published in Opt. Lett. 2019, and (ii) J. Lee et al. "High-purity reflective color filters based on thin film cavities embedded with an ultrathin Ge₂Sb₂Te₅ absorption layer" published in Nanoscale Adv. 2020. While the authors have made a quantitative comparison of each optimized structure through an optimization algorithm, a direct comparison with the aforementioned studies would provide a better understanding of the contributions of this work to the field. Therefore, I would appreciate it if the authors could include a discussion of how their results compare to those of the cited studies. This would provide additional insights into the novelty and significance of their proposed platform and its potential impact in the field of optical coatings.

We thank the reviewer for their positive response to our revised manuscript, and for bringing these two benchmark structural coloring studies to our attention. We agree that including a comparison of the structural coloring capabilities of these platforms will significantly increase the comprehensiveness of our comparison with FROCs. These two studies have been compared and are included in **Figure 5a** of the manuscript, as shown below. In both cases FROCs outperform the other platforms in terms of total color gamut coverage. Similarly, FROCs outperform the platform shown in Lee 2020 [2] in purity across the gamut and show similar purities to those demonstrated in C. Ji et. al., 2019 [1]. It should also be noted that in reference [1] the described

system is only shown capable of high purity red (~600 nm) color, and does not represent a comprehensive structural coloring platform. This lack of color gamut coverage is likely due to the mechanism adopted by the authors where high order Fabry-Perot resonance create a broad absorption band in the entire visible range except for the red color. In addition, they added an AR coating to make the reflection steep. However, this approach cannot be adopted straightforwardly for green colors as an example.

Figure 5| Accessible color gamut, brightness and overall performance of FROCs: (a) CIE 1931 chromaticity diagram comparing FROCs and Silica-capped FROCs with Fabry-Perot cavities and other reported thin-film based coloring approaches from the recent literature at normal incidence. (b) Measured reflectance from different FROCs showing a peak reflectance ranging from 0.63-0.85. The high reflectance corresponds to high brightness, a desirable property in structural colors. (c) Comparison between the coloring performance of FROC vs. other structural coloring platforms.

Additionally, we have included a table in the supplementary information (**Table S12**) comparing FROCs with each of the structural coloring platforms shown in **Figure 5a**, in terms of total gamut coverage and maximum purity at two incidence angles (0 deg and 55 deg). We believe this table will facilitate an easier comparison between FROCs and the benchmark studies, including those proposed by the reviewer.

Comparison of Thin-Film Structural Coloring Platforms:

Comparison of Thin-Film Structural Coloring Platforms in the Current Literature				
S-C Platform	Gamut Coverage: 0 deg	Gamut Coverage: 55 deg	Max. Purity: 0 deg	Max. Purity: 55 deg
FROC	44%	62%	95%	99%
SiO2 Capped FROC	61%	62%	98%	99%
MDM (All Layers)	32%	32%	95%	97%
MDM (Opaque Base)	4%	4%	60%	59%
Zhao 2019 [1]	15%	14%	78%	76%
Ghobadi 2019 [2]	4%	5%*	64%	71%*
Ghobadi 2019 [3]	20%	12%*	75%	58%*
Park 2021 [4]	32%	36%	93%	95%
Lee 2020 [5]	28%	—	78%	—
Ji 2019 [6]	—	—	83%	59%*

* Angle of incidence is 60 deg

Table S12| Comparison of Thin-Film Structural Coloring Platforms: Total gamut coverage and maximum purity of FROCs compared with thin-film structural coloring platforms for 0 deg and 55 deg angles of incidence. The best performing system in each category is highlighted in blue.

References

- [1] Lee, Junho, Jaeyong Kim, and Myeongkyu Lee. High-purity reflective color filters based on thin film cavities embedded with an ultrathin Ge₂Sb₂Te₅ absorption layer. *Nanoscale Advances* **2**, 10, 4930-4937 (2020).
- [2] Ji, Chengang, Kyu-Tae Lee, and L. Jay Guo. High-color-purity, angle-invariant, and bidirectional structural colors based on higher-order resonances. *Optics Letters* **44**, 1, 86-89 (2019).

REVIEWERS' COMMENTS

Reviewer #1 (Remarks to the Author):

Upon careful review of the revised manuscript and response letter, I believe that the authors have adequately addressed the concerns raised in my previous review.

Response To Decision Letter

Manuscript #: NCOMMS-22-38037

Title: Fano Resonant Optical coatings platform for Full Gamut and High Purity Structural Colors

Authors: M. ElKabbash et. al

Responses in blue text

Reviewer #1:

Upon careful review of the revised manuscript and response letter, I believe that the authors have adequately addressed the concerns raised in my previous review.

We thank the reviewer for their time and effort and for accepting our replies.